# Entanglement Phase Transitions in Non-Hermitian Kitaev Chains

**DOI:** 10.3390/e26030272

**Published:** 2024-03-20

**Authors:** Longwen Zhou

**Affiliations:** 1College of Physics and Optoelectronic Engineering, Ocean University of China, Qingdao 266100, China; zhoulw13@u.nus.edu; 2Key Laboratory of Optics and Optoelectronics, Qingdao 266100, China; 3Engineering Research Center of Advanced Marine Physical Instruments and Equationipment of MOE, Qingdao 266100, China

**Keywords:** entanglement entropy, quantum phase transition, non-Hermitian topological phase

## Abstract

The intricate interplay between unitary evolution and projective measurements could induce entanglement phase transitions in the nonequilibrium dynamics of quantum many-particle systems. In this work, we uncover loss-induced entanglement transitions in non-Hermitian topological superconductors. In prototypical Kitaev chains with onsite particle losses and varying hopping and pairing ranges, the bipartite entanglement entropy of steady states is found to scale logarithmically versus the system size in topologically nontrivial phases and become independent of the system size in the trivial phase. Notably, the scaling coefficients of log-law entangled phases are distinguishable when the underlying system resides in different topological phases. Log-law to log-law and log-law to area-law entanglement phase transitions are further identified when the system switches between different topological phases and goes from a topologically nontrivial to a trivial phase, respectively. These findings not only establish the relationships among spectral, topological and entanglement properties in a class of non-Hermitian topological superconductors but also provide an efficient means to dynamically reveal their distinctive topological features.

## 1. Introduction

The entanglement dynamics of open quantum many-body systems undergoing nonunitary evolution have attracted significant attention in recent years [1,2,3]. An intriguing phenomena that could emerge in such contexts is the measurement-induced entanglement transition [4,5,6,7,8,9,10]. It describes a nonequilibrium phase transition of entanglement structures following a quantum quench. In usual situations, measurement-induced entanglement phase transitions originate from the competition between unitary dynamics and quantum measurements. With the increase in measurement rates, the bipartite entanglement entropy (EE) of nonequilibrium steady states could undergo a transition from a volume-law to an area-law scaling versus the system size. Despite great theoretical efforts [11,12,13,14,15,16,17,18,19,20,21,22,23,24,25,26,27,28,29,30,31,32,33,34,35,36,37,38,39,40,41,42,43,44], measurement-induced entanglement transitions have also been explored experimentally in setups including trapped ions and superconducting qubits [45,46,47], offering further insights for understanding quantum information dynamics and simulating quantum many-body systems.

Open quantum systems described by non-Hermitian Hamiltonians constitute an important context for exploring entanglement phase transitions. Various types of non-Hermiticity-induced entanglement transitions have been identified in gapped or critical non-Hermitian systems made up of lattice fermions and quantum spin chains [48,49,50,51,52,53,54,55,56,57,58,59,60,61,62,63,64]. In a non-Hermitian system with spatial nonreciprocity, the emergence of non-Hermitian skin effects was found to accompany the transition from a volume-law entangled to an area-law entangled phase in one spatial dimension [52]. The development of a dissipation gap in the energy spectrum of a non-Hermitian Hamiltonian was also found to yield a volume-to-area law entanglement phase transition in free-fermion chains [53]. Moreover, in non-Hermitian systems with spatially uniform or quasiperiodic randomness [55,56,57,58], entanglement phase transitions beyond the conventional volume-law to area-law scheme could emerge due to the interplay between disorder and non-Hermitian effects. In addition, alternated and re-entrant entanglement transitions may be engineered and controlled by time-periodic driving fields in non-Hermitian Floquet systems [59].

In this work, we continue the study of entanglement phase transitions in non-Hermitian systems. We focus on one-dimensional (1D) topological superconductors with onsite particle losses, which have been found to possess rich topological phases and dynamical phase transitions [65,66,67,68,69,70,71,72,73]. In Section 2, we introduce our model and outline the approaches of characterizing its spectrum, topological properties and entanglement dynamics. In Section 3, we explore entanglement phase transitions in representative non-Hermitian Kitaev chains with varying ranges of single-particle hopping and superconducting pairing terms. All-round connections are established among the spectral structures, topological transitions and entanglement phase transitions in the considered system. Remarkably, each loss-induced transition between topologically distinct superconducting phases is found to go hand-in-hand with a transition in the scaling law of steady-state EE, i.e., an entanglement phase transition. In Section 4, we summarize our results and discuss potential future directions. Some further calculation details about the non-Hermitian Hamiltonian and entanglement dynamics are provided in the Appendix A and Appendix B.

## 2. Model and Methods


We start by considering a generalized Kitaev chain [74] with onsite particle losses, whose Hamiltonian can be expressed as
(1)$$\hat{H}=\frac{1}{2}\sum_{n}\left[\mu \left(2{\hat{c}}_{n}^{†}{\hat{c}}_{n}-1\right)+\sum_{r}\left(J_{r}{\hat{c}}_{n}^{†}{\hat{c}}_{n+r}+\Delta_{r}{\hat{c}}_{n}{\hat{c}}_{n+r}+H.c.\right)\right].$$Here, H.c. denotes Hermitian conjugation, n∈Z represents the lattice index (with the lattice constant a=1), and r∈Z controls the ranges of hopping and pairing terms. ${\hat{c}}_{n}^{†}$ (${\hat{c}}_{n}$) creates (annihilates) a spin-polarized fermion on the *n*th lattice site. The chemical potential μ=u−iv is chosen to be complex, where u=Reμ, and v=−Imμ>0 describing the loss rate. $J_{r}$ and $\Delta_{r}$ denote hopping and pairing amplitudes of fermions over *r* lattice sites [r=1 for the nearest-neighbor (NN) coupling, r=2 for the next-nearest-neighbor (NNN) coupling, etc.]. The presence of long-range hopping and pairing terms beyond nearest-neighbor (r>1) allows the system to possess superconducting phases with large topological invariants [72]. We will refer to the system described by $\hat{H}$ in Equation (Equation 1) with v>0 as the lossy Kitaev chain (LKC). Note in passing that the effective non-Hermitian Hamiltonian in Equation (Equation 1) can be viewed as emerging from a stochastic evolution followed by post-selecting null measurement outcomes. Some further derivation details of $\hat{H}$ are provided in the Appendix A.

Taking the periodic boundary condition (PBC) ${\hat{c}}_{n}={\hat{c}}_{n+L}$ and applying the Fourier transformation ${\hat{c}}_{n}=\frac{1}{\sqrt{L}}\sum_{k}e^{ikn}{\hat{c}}_{k}$, we can express $\hat{H}$ in the momentum space as $\hat{H}=\frac{1}{2}\sum_{k}{\hat{\Psi }}_{k}^{†}H\left(k\right){\hat{\Psi }}_{k}$, where *L* is the length of lattice, k∈[−π,π) is the quasimomentum, ${\hat{\Psi }}_{k}^{†}=\left({\hat{c}}_{k}^{†},{\hat{c}}_{-k}\right)$ is the Nambu spinor operator, and
(2)$$H\left(k\right)=h_{y}\left(k\right)\sigma_{y}+h_{z}\left(k\right)\sigma_{z},$$
(3)$$h_{y}\left(k\right)=\sum_{r}\Delta_{r}\sin\left(kr\right),$$
(4)$$h_{z}\left(k\right)=\mu +\sum_{r}J_{r}\cos\left(kr\right).$$Here, $\sigma_{x,y,z}$ are Pauli matrices in their usual representations. By diagonalizing H(k), we can find the spectrum of the system as $E_{\pm }\left(k\right)=\pm E\left(k\right)$, where
(5)$$E\left(k\right)=\sqrt{h_{y}^{2}\left(k\right)+h_{z}^{2}\left(k\right)}.$$With the loss rate v>0 in the complex chemical potential μ=u−iv, E(k) becomes complex in general. Meanwhile, H(k) possesses the chiral symmetry $S=\sigma_{x}$, in the sense that SH(k)S=−H(k). The topological phases of the system could then be characterized by a winding number *w* [72], which is defined as
(6)$$w=\frac{1}{2\pi }\int_{-\pi }^{\pi }dk\partial_{k}\phi \left(k\right),$$
where $\phi \left(k\right)=\arctan[h_{y}\left(k\right)/h_{z}\left(k\right)]$. The value of 2πw is equal to the accumulated winding angle ϕ(k) over the first Brillouin zone (BZ) of *k*. As the imaginary part of ϕ(k) has no winding [75], *w* can only take real values. Moreover, when the energy spectrum is gapped, *w* will take an integer (a half-integer) quantized value if the trajectory of vector $\left[h_{y}\left(k\right),h_{z}\left(k\right)\right]$ encircles an even (odd) number of exceptional points (EPs) of H(k) following the change of *k* from −π to π [72]. The closing and reopening of spectrum gaps will also accompany the quantized (or half quantized) changes of *w*. Therefore, we can characterize the non-Hermitian topological phases and topological phase transitions of our LKC by the winding number *w* in Equation (Equation 6).

In this work, we focus on the loss-induced entanglement phase transitions and their connections with topological phases in non-Hermitian Kitaev chains. To deal with the entanglement dynamics, we first prepare our system in a certain initial state |Ψ(0)〉 at half-filling and evolve it over a time duration *t* according to the Hamiltonian $\hat{H}$ [Equation (Equation 1)]. The resulting state of the system then takes the form (ℏ=1)
(7)$$|\Psi \left(t\right)\rangle =\frac{e^{-i\hat{H}t}|\Psi \left(0\right)\rangle }{\left|\right|e^{-i\hat{H}t}\left|\Psi \left(0\right)\rangle |\right|},$$
where the normalization factor $\left|\right|e^{-i\hat{H}t}\left|\Psi \left(0\right)\rangle |\right|$ could arise after taking the no-click limit of a monitored evolution [52,53]. With the normalized state |Ψ(t)〉, we could obtain the single-particle correlator C(t) of the system in position representation. Due to the translational symmetry of our system, the matrix elements $C_{m,n}\left(t\right)$ (m,n=1,…,L) of C(t) in real-space can be further computed by performing the Fourier transformation of its related generator in *k*-space, i.e.,
(8)$$C_{m,n}\left(t\right)=\frac{1}{L}\sum_{k}e^{ik(m-n)}C_{k}\left(t\right),$$
where the 2×2 matrix $C_{k}\left(t\right)$ can be obtained as
(9)$$C_{k}\left(t\right)=\frac{1}{2}\left(\begin{array}{cc} 1+{\langle \sigma_{z}\rangle }_{t} & {\langle \sigma_{x}\rangle }_{t}+i{\langle \sigma_{y}\rangle }_{t}\\ {\langle \sigma_{x}\rangle }_{t}-i{\langle \sigma_{y}\rangle }_{t} & 1-{\langle \sigma_{z}\rangle }_{t} \end{array}\right).$$The average ${\langle \sigma_{j}\rangle }_{t}$ (j=x,y,z) is taken over the normalized state $|\psi_{k}\left(t\right)\rangle$, which is evolved by $e^{-iH(k)t}$ from the initial state vector $|\psi_{k}\left(0\right)\rangle$ at each quasimomentum *k* (see Appendix B for more details).

Finally, to obtain the bipartite EE, we decompose our 1D chain of length *L* into two spatially connected subsystems A and B, whose number of lattice sites are *l* and L−l, respectively. Restricting the lattice indices *m* and *n* to the A segment gives us the correlation matrix $C_{m,n}^{A}\left(t\right)$ (m,n=1,…,l) of subsystem A. Then, according to the relation between single-particle correlation matrix and bipartite EE of a Gaussian state [76], we can express the EE between the two subsystems A and B as
(10)$$S\left(t\right)=-\sum_{j=1}^{l}\left[\zeta_{j}\ln\zeta_{j}+\left(1-\zeta_{j}\right)\ln\left(1-\zeta_{j}\right)\right],$$
where $\left\{\zeta_{j}|j=1,\ldots ,l\right\}$ are the eigenvalues of the l×l correlation matrix $C^{A}\left(t\right)$. In numerical calculations, we can obtain S(t) efficiently at any time *t* through Equations (Equation 7)–(Equation 9) together with the diagonalization of $C^{A}\left(t\right)$. In the long-time limit, S(t) will reach a stationary value $S\left(L,l\right)\equiv \lim_{t\to \infty }S\left(t\right)$, which may depend on the total system size, the subsystem size and other system parameters. We can numerically find the bipartite EE S(L,l) of steady states by considering an evolution time duration t∈[0,T] that is long enough, so that any variations of S(t) over time are negligible for t≥T.

In the next section, we explore entanglement phase transitions in two representative LKC models with distinct topological properties. We will see clear changes in the scaling behaviors of steady-state EE when the considered system undergoes topological phase transitions. Moreover, qualitatively different scaling laws of EE in steady states will be uncovered when the system resides in topological and trivial non-Hermitian superconducting phases, respectively.

## 3. Results

In this section, we investigate entanglement phase transitions in two non-Hermitian Kitaev chains with different ranges of hopping and pairing terms. We first reveal the spectrum and topological properties of each model. This is followed by the demonstration of their entanglement dynamics and the related scaling laws of steady-state EE. Finally, we establish the entanglement phase diagrams and further unveil their connections with the topological phases and transitions for both models.

### 3.1. LKC with Nearest-Neighbor Hopping and Pairing

We start with a “minimal” model of LKC by restricting the hopping and pairing terms to NN sites. Referring to Equation (Equation 1), we choose r=1, let $\left(J_{1},\Delta_{1}\right)=\left(J,\Delta \right)$, and set the chemical potential μ=u−iv. The resulting system is described by the Hamiltonian
(11)$${\hat{H}}_{1}=\frac{1}{2}\sum_{n}\left[\mu \left({\hat{c}}_{n}^{†}{\hat{c}}_{n}-1/2\right)+J{\hat{c}}_{n}^{†}{\hat{c}}_{n+1}+\Delta {\hat{c}}_{n}{\hat{c}}_{n+1}+H.c.\right].$$It has been identified that under PBC, the two energy bands $E_{\pm }\left(k\right)=\pm E\left(k\right)$ [Equation (Equation 5)] of this NN LKC model become gapless at E=0 when the system parameters satisfy the condition $u^{2}/J^{2}+v^{2}/\Delta^{2}=1$ [72]. When $u^{2}/J^{2}+v^{2}/\Delta^{2}<1$, there is a line gap lying along the ReE axis between the two bands ±E(k) on the complex energy plane. Within the gap, a pair of degenerate Majorana zero modes can be found at the edges of the chain under the open boundary condition (OBC), and the system is topologically nontrivial in this case [72]. When $u^{2}/J^{2}+v^{2}/\Delta^{2}>1$, a line gap in the energy spectrum is opened along the ImE axis, and there are no localized Majorana modes with zero energy at system edges under the OBC. In this case, the system becomes topologically trivial and possesses a trivial dissipation gap [72].

In Figure 1, we present typical energy spectra on the complex plane and the topological phase diagram of the NN LKC model [Equation (Equation 11)]. In Figure 1a, the phase diagram is obtained by computing the topological winding number *w* [Equation (Equation 6)] at different system parameters (u,v). The results confirm that the NN LKC indeed belongs to the topologically nontrivial (trivial) phase with w=1 (w=0) when $u^{2}/J^{2}+v^{2}/\Delta^{2}<1$ (>1), with the phase boundary $u^{2}/J^{2}+v^{2}/\Delta^{2}=1$ given by the black solid line between the yellow and green regions in Figure 1a. In Figure 1b,c, we illustrate the complex spectrum of the system in the topological and trivial phases, respectively, under the OBC. A real energy gap carrying a pair of Majorana edge modes at E=0 is observed in the topological phase, while an imaginary energy gap with no Majorana zero modes is found in the trivial phase, which validates our theoretical predictions [72].

With the above knowledge on the spectral and topological properties of the NN LKC, it would be interesting to check whether the dynamics and scaling laws of its EE could show distinct behaviors in different topological phases. Following Equations (Equation 7)–(Equation 10), we compute the evolution of bipartite EE S(t) of the NN LKC under PBC with a large system size *L*. A set of representative results versus different sizes *l* of the subsystem A and at different loss rates are shown in Figure 2. In all of the calculations, we choose the initial state at different quasimomenta as
(12)$$|\psi_{k}\left(0\right)\rangle =\frac{1}{\sqrt{2}}\left(\begin{array}{c} 1\\ e^{ik/2} \end{array}\right),$$
where k=−π,−π+2π/L,…,π−2π/L. Other forms of pure and non-stationary initial states generate consistent results regarding the entanglement dynamics. We find that when the system parameters satisfy $u^{2}/J^{2}+v^{2}/\Delta^{2}<1$, the bipartite EE will first experience a transient time window with a non-monotonous behavior in time, and finally evolving to a stationary value in the late time regime. After the steady state is reached, the EE raises monotonically with the increase in subsystem size *l* (solid lines in Figure 2). On the contrary, in the parameter regime with $u^{2}/J^{2}+v^{2}/\Delta^{2}>1$, the bipartite EE will converge to the same steady-state value for any subsystem size *l* after a transient time window, which implies that $\lim_{t\to \infty }S\left(t\right)∼l^{0}$ in this region (dashed lines in Figure 2). In summary, these observations suggest that in the NN LKC, the scaling laws of steady-state EE vs. the subsystem size could indeed be qualitatively different in the topological phase with a real energy gap and the trivial phase with an imaginary energy gap. A loss-driven entanglement phase transition may then occur and go hand-in-hand with the topological phase transition of the system.

To further confirm the existence of different entangling phases in the NN LKC, we present in Figure 3 the steady-state EE S(L,l) versus the subsystem size *l* and loss rate *v*. The steady-state is reached by evolving the initial state |Ψ(0)〉 according to Equation (Equation 7) [with $\hat{H}={\hat{H}}_{1}$ in Equation (Equation 11)] over a long time duration, which is set to T=2000 in our calculations. In Figure 3a, we identify the scaling-law of steady-state EE as S(L,l)∼ln[sin(πl/L)] for $u^{2}/J^{2}+v^{2}/\Delta^{2}<1$ and $S\left(L,l\right)∼l^{0}$ for $u^{2}/J^{2}+v^{2}/\Delta^{2}>1$, respectively, with v≠0. In the thermodynamic limit (L→∞), we obtain the following scaling laws for the bipartite EE of steady states at half-filling, i.e.,
(13)$$S\left(L,l\right)∼\left\{\begin{array}{cc} \ln l, & v\in (|\Delta |\sqrt{1-u^{2}/J^{2}},\infty ),\\ l^{0}, & v\in (0,|\Delta |\sqrt{1-u^{2}/J^{2}}). \end{array}\right.$$Therefore, the EE satisfies a log-law (an area-law) vs. the subsystem size in the weak (strong) dissipation regime of the system. These two distinct entangling phases are clearly illustrated in the regions with v<0.6 and v>0.6 in Figure 3b. Remarkably, the separation point [red dashed line at v=0.6 in Figure 3b] between these two entangling phases is precisely coincident with the topological transition point of the NN LKC at $u^{2}/J^{2}+v^{2}/\Delta^{2}=1$. There should thus be a transition from a log-law entangled topological superconducting phase to an area-law entangled trivial phase in the NN LKC with the increase in the loss rate *v*.

Based on the above analysis, we could identify the entanglement phase transition and establish the entanglement phase diagram of the NN LKC, as reported in Figure 4. To reveal the change in scaling behaviors in the steady-state EE, we fit S(L,l) with the function gln[sin(πl/L)] in different parameter regions and extract the coefficient *g* as the gradient of the associated scaling laws. In Figure 4a, we take J=Δ=1 and the real part of chemical potential u=0.8. Referring to the topological phase boundary, $v=\left|\Delta \right|\sqrt{1-u^{2}/J^{2}}$, we find the topological transition point to be v=0.6 in this case. It is clear that we have a finite *g* when v<0.6 and a vanishing *g* for v>0.6 in Figure 4a, which verifies the presence of two different entangling phases (log-law vs. area-law) and an entanglement phase transition driven by the change in loss rate *v* in the NN LKC.

Finally, we present the gradient *g* extracted from the fitting S(L,l)∼gln[sin(πl/L)] at different (u,v) in Figure 4b, generating the entanglement phase diagram of the system. We observe two different regions with distinct entanglement features (g>0 vs. g=0), which are separated by the boundary line $v=\left|\Delta \right|\sqrt{1-u^{2}/J^{2}}$ for v≠0 and |u|≤|J|. A direct comparison with the topological phase diagram in Figure 1a leads to the conclusion that for the NN LKC, the bipartite EE of steady states follows the log-law scaling vs. the subsystem size in the topological phase (with winding number w=1) and becomes independent of the subsystem size in the trivial phase (with w=0). At v≠0, a log-law to area-law entanglement phase transition could happen following the transition of the system from a topologically nontrivial to a trivial phase. This entanglement transition could thus offer a unique dynamical probe to topological phase transitions in non-Hermitian Kitaev chains.

A possible mechanism behind the loss-driven entanglement transition in the NN LKC is as follows. With v>0, the particles tend to populate the energy levels with positive imaginary parts after being evolved over a long time duration. In the regime with $v>|\Delta |\sqrt{1-u^{2}/J^{2}}$, such energy levels are separated apart from those with negative imaginary parts by a dissipation gap [see Figure 1c]. The particle distribution of late-time steady state then mimics a 1D normal band insulator at half filling, whose bipartite EE is expected to follow an area-law scaling versus the system size. In the regime with $v<|\Delta |\sqrt{1-u^{2}/J^{2}}$, there is no dissipation gap between the energy levels with positive and negative imaginary parts [see Figure 1b]. After a long evolution time, the steady-state population tends to form an effective Fermi surface at E=0 along the ReE axis, which has two crossing points with the bulk spectrum. Therefore, the particle distribution of steady state is close to a critical metallic phase in 1D systems, whose bipartite EE is expected to scale logarithmically with respect to the system size [77,78,79].

In the next subsection, we investigate entanglement transitions in an LKC with NNN hopping and pairing terms, whose topological phases could possess larger winding numbers. We will see that the connections between topological and entanglement phase transitions found in this subsection could be extended to more general situations for non-Hermitian topological superconductors.

### 3.2. LKC with Next-Nearest-Neighbor Hopping and Pairing

We now consider an LKC with second-neighbor hopping and pairing terms. Referring to Equation (Equation 1), we set the coupling range r=2, and the resulting Hamiltonian of the system reads
(14)$${\hat{H}}_{2}=\frac{1}{2}\sum_{n}\left[\mu \left(2{\hat{c}}_{n}^{†}{\hat{c}}_{n}-1\right)+\sum_{r=1,2}\left(J_{r}{\hat{c}}_{n}^{†}{\hat{c}}_{n+r}+\Delta_{r}{\hat{c}}_{n}{\hat{c}}_{n+r}+H.c.\right)\right].$$The non-Hermitian effect is again introduced by setting the chemical potential μ=u−iv with v>0. In momentum space, it can be shown that there are four possible critical quasimomenta $\pm k_{0}^{\pm }=\pm \arccos\left\{\left[-J_{1}\pm \sqrt{J_{1}^{2}+8J_{2}\left(J_{2}-u\right)}\right]/\left(4J_{2}\right)\right\}$, where the two bulk energy bands of ${\hat{H}}_{2}$ could touch with each other at E=0 [72]. The topological phase boundaries of the system in parameter space can then be obtained by solving the equations $\sin\left(k_{0}^{\pm }\right)\left(\Delta_{1}+2\Delta_{2}\cos k_{0}^{\pm }\right)=\pm v$ [72]. In Figure 5a, we plot these phase boundaries as black lines in the u−v plane and compute the topological winding number *w* [Equation (Equation 6)] in different parameter regions. The results show that there are three different topological phases with winding numbers w=0,1,2. The presence of NNN hopping and coupling terms allows us to obtain a non-Hermitian topological superconducting phase with a larger winding number w=2.

In the topological phase with w=2, the bulk energy spectrum of the system is gapped along the ReE axis, and there are two pairs of Majorana edge modes at E=0 under the OBC [see Figure 5b]. With the increase in *v*, the system could undergo a phase transition through level crossings at E=0, after which it enters another topological phase with w=1. In this phase, the bulk spectrum of the system holds a line energy gap along a certain angle θ∈(0,π/2) with respect to the ReE axis, and a single pair of Majorana edge modes can be found inside the gap at E=0 under the OBC [see Figure 5c]. Note in passing that this non-Hermitian topological superconducting phase is different from the w=1 phase of NN LKC [see Figure 1], whose spectral gap lies instead along the ReE axis. With the further increase in *v*, the system would experience a second phase transition and finally entering a trivial phase with w=0. In this phase, the spectrum of the system develops a line gap along the ImE axis, and there are no zero-energy Majorana edge modes within the gap under the OBC [see Figure 5d].

After unveiling the rich spectral and topological features of the NNN LKC, we are ready to explore its entanglement dynamics. Referring again to Equations (Equation 7)–(Equation 10), we can find the evolution of bipartite EE S(t) of the NNN LKC with a large system size *L* under PBC. Exemplary results vs. different sizes *l* of the subsystem A and different loss rates are presented in Figure 6. We have also considered a half-filled system and used Equation (Equation 12) as the initial state at different quasimomenta throughout the calculations. The results show that in the case with a large loss rate (v=2.5 in Figure 6), the EE does not change with the increase in subsystem size *l* after a long evolution time (dotted lines in Figure 6), which implies an area-law scaling of the steady-state EE vs. *l*. This case corresponds to the spectrum of NNN LKC in Figure 5d, where there exists a dissipation gap, and the system resides in a topologically trivial phase. In the situation with a small loss rate (v=0.1 in Figure 6), the EE grows monotonically with the increase in the subsystem size *l* after a long-time evolution (solid lines in Figure 6). It corresponds to the case with a line energy gap along the ReE axis in the spectrum [Figure 5b], and the system resides in a topological phase with winding number w=2. When the loss rate takes an intermediate value (v=0.8 in Figure 6), the EE again rises with the increase in the subsystem size *l* in the late time regime. Nevertheless, its growth rate vs. *l* is smaller compared to the case of v=0.1. The system also belongs to a topological phase with a smaller winding number w=1 and possessing a gapped spectrum as shown in Figure 5c. These observations suggest that for the NNN LKC, the bipartite EE in steady states may possess different scaling properties when the system belongs to distinct topological superconducting phases with distinguishable spectral features.

To further decode the scaling laws of bipartite EE in steady states, we consider the evolution of the system over a long time duration T=2000 and obtain the final values of EE S(L,l) at different subsystem sizes *l* and loss rates, as shown in Figure 7. The methodology of computing S(L,l) here is in parallel with that employed in Section 3.1. We find that in the weak and intermediate dissipation regions [v=0.1,0.5 and v=1.1,1.5 in Figure 7a], the EE S(L,l) is proportional to ln[sin(πl/L)] for a fixed system size *L*. In the limit of large *L*, this relation reduces to the log-law scaling S∼lnl. In the strong dissipation region [v=2.1,2.5 in Figure 7a], the steady-state EE becomes independent of *l*, leading to an area-law entangled phase with $S∼l^{0}$. More precisely, under the condition v>0, we numerically find the following three parameter regions in which the bipartite EE of steady states show distinguishable scaling behaviors in the limit L→∞, i.e.,
(15)$$S\left(L,l\right)∼\left\{\begin{array}{cc} g\ln l, & v<\min|h_{y}\left(k_{0}^{\pm }\right)|,\\ g^{\prime }\ln l, & \min|h_{y}\left(k_{0}^{\pm }\right)\left|<v<\max\right|h_{y}\left(k_{0}^{\pm }\right)|,\\ l^{0}, & v>\max|h_{y}\left(k_{0}^{\pm }\right)|. \end{array}\right.$$Here, the coefficients *g* and $g^{\prime }$ both depend on the system parameters, and we always have $g>g^{\prime }$. The expression of $h_{y}\left(k\right)$ is given by Equation (Equation 3) after setting r=1,2. $k_{0}^{\pm }\in \left[-\pi ,\pi \right)$ are the critical quasimomenta where the energy bands could touch when a topological phase transition happens [72]. The three distinct scaling regions of S(L,l) are clearly visible from the EE vs. loss rate *v* in Figure 7b. Notably, the separation points between these entangling phases tend to be identical to the topological transition points of the NNN LKC [72]. Despite a transition from a log-law entangled topological phase to an area-law entangled trivial phase, there should also be a possible entanglement transition between two non-Hermitian topological superconducting phases with winding numbers w=2 and w=1 following the variation in the loss rate *v*.

To confirm the presence of entanglement transitions in the NNN LKC and build their connections with topological phase transitions, we perform numerical fitting for the bipartite EE of steady states as S(L,l)∼gln[sin(πl/L)] and extract the coefficients *g* at different system parameters. The results are shown in Figure 8. The two red dashed lines in Figure 8a correspond to the loss rates where topological phase transitions happen in the system [72]. We see that they separate the configuration of *g* into three distinguishable regions. In the left and middle regions (with weak and intermediate dissipation), the values of *g* are finite and decrease gradually with the increase in *v*. The system then belongs to log-law entangled phases in these domains. In the right region, the gradient *g* becomes pinned to zero, which implies that the system has entered an area-law entangled phase with $S\left(L,l\right)∼l^{0}$. It is noteworthy that the derivatives of *g* with respect to the loss rate *v* undergo two discontinuous changes at the topological phase transition points. Therefore, the loss-induced topological transitions in the NNN LKC also accompany entanglement phase transitions characterized by different scaling laws in the bipartite EE of steady states.

Finally, we present the scaling coefficients *g* of S(L,l) vs. the real and imaginary parts of chemical potential μ in Figure 8b, which forms the entanglement phase diagram of the system. In comparison with the topological phase diagram in Figure 5a, we conclude that the topological phases of NNN LKC with different winding numbers w=2, 1 and 0 indeed exhibit different scaling laws in the EE vs. system sizes. Both the two topologically nontrivial phases are log-law entangled, whereas the topologically trivial phase is area-law entangled. Each topological phase transition further goes hand-in-hand with an entanglement phase transition.

The physical mechanism behind these loss-driven entanglement transitions is similar to that discussed for the NN LKC in Section 3.1. Starting with an initial state at half-filling, the final population of the steady state tends to fill the energy levels with positive imaginary parts after a long evolution time. The interface between filled and empty states then possesses four, two and no crossing points with the bulk energy spectrum of ${\hat{H}}_{2}$ in the topological phases with winding numbers w=2, 1 and 0, respectively, (see Figure 5), yielding two critical-type, log-law entangled phases and a gapped, area-law entangled phase. The entanglement phase transitions identified here may also provide us with an alternative strategy to dynamically probe and distinguish non-Hermitian topological superconducting phases with large winding numbers.

## 4. Conclusions

In this work, we unveiled entanglement phase transitions in 1D non-Hermitian topological superconductors. Our investigation focused on Kitaev chains exhibiting onsite particle losses, revealing distinct scaling laws of EE corresponding to different non-Hermitian superconducting phases. Specifically, the bipartite EE of steady states demonstrated an area-law scaling in the topologically trivial phase and a log-law scaling in the nontrivial phase. Notably, the scaling coefficients of the log-law become distinguishable when the system is situated in superconducting phases with distinct topological winding numbers. Furthermore, we identified entanglement phase transitions coinciding with topological phase transitions in non-Hermitian Kitaev chains at identical system parameters. Our study established generic connections among spectral, topological, and entanglement phase transitions in two distinct lossy Kitaev chains with varying hopping and pairing ranges. These findings not only uncovered the richness and unique characteristics of entanglement transitions in a class of non-Hermitian systems, but also introduced an efficient dynamical probe for detecting and distinguishing different non-Hermitian superconducting phases with diverse spectral and topological properties.

In future research, exploring entanglement phase transitions in non-Hermitian topological superconductors with disorder, subject to time-periodic driving, and extending the investigation to higher spatial dimensions holds considerable interest. The critical properties of entanglement transitions in non-Hermitian systems and their responses to many-body interactions deserve more thorough explorations. Additionally, the experimental realization of non-Hermitian Kitaev chains and the detection of entanglement and topological phase transitions within these systems offer intriguing directions for future studies.

## Figures and Tables

**Figure 1 entropy-26-00272-f001:**
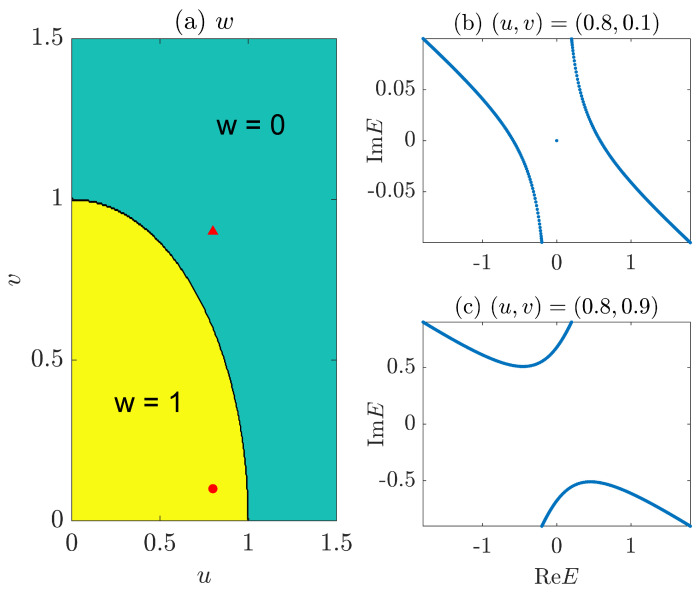
Topological phase diagram and typical spectra of the LKC with NN hopping and pairing. (**a**) shows the winding number *w* vs. the real and imaginary parts of chemical potential *u* and *v*. The yellow and green regions have w=1 and w=0, respectively. The red solid dot in (**a**) resides at (u,v)=(0.8,0.1), with the associated spectrum of ${\hat{H}}_{1}$ shown in (**b**). The red solid triangle of (**a**) is located at (u,v)=(0.8,0.9), and the associated spectrum of ${\hat{H}}_{1}$ is given in (**c**) on the complex energy plane. Other system parameters are J=Δ=1 for all panels.

**Figure 2 entropy-26-00272-f002:**
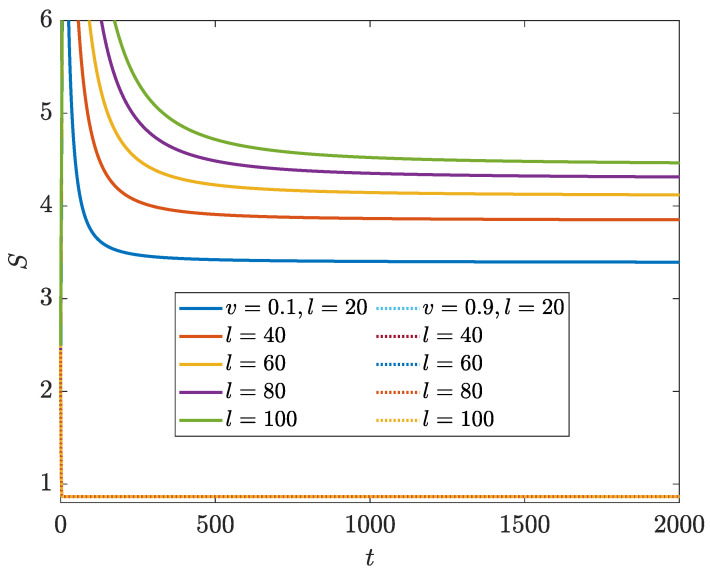
Bipartite EE vs. time *t* for the LKC with NN hopping and pairing for the loss rate v=0.1 (solid lines) and 0.9 (dotted lines). Other system parameters are J=Δ=1 and u=0.8. *l* denotes the subsystem size and the total lattice size is $L=2\times 10^{4}$.

**Figure 3 entropy-26-00272-f003:**
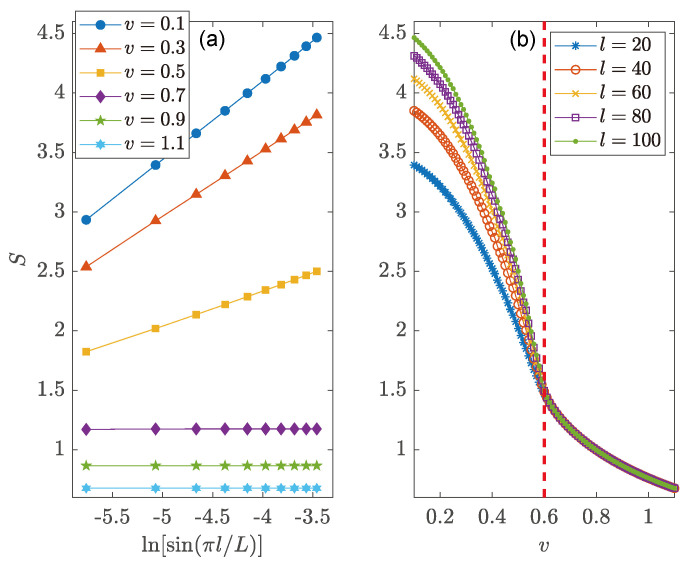
Bipartite EE of steady states vs. (**a**) the subsystem size *l*, and (**b**) the loss rate *v* for the LKC with NN hopping and pairing. Other system parameters are J=Δ=1 and u=0.8 for both panels. The lattice size of the whole system is $L=2\times 10^{4}$. The vertical dashed line in (**b**) highlights the phase transition point at v=0.6.

**Figure 4 entropy-26-00272-f004:**
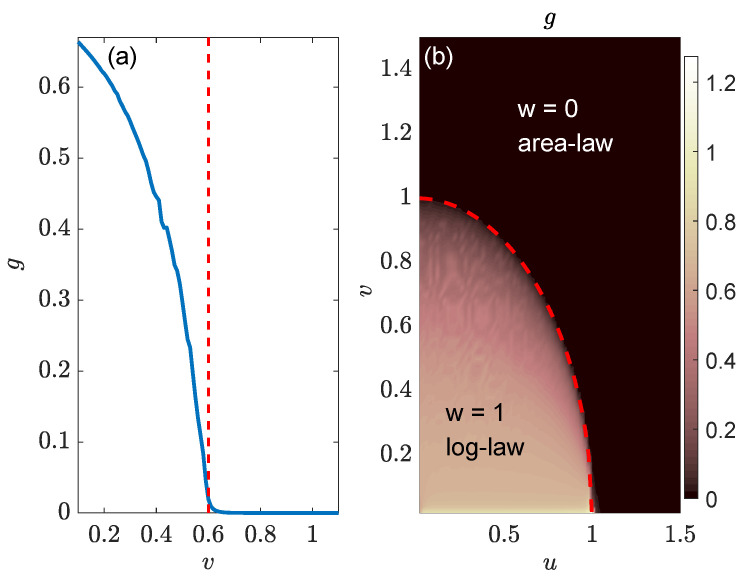
Entanglement phase transitions in the LKC with NN hopping and pairing. System parameters are J=Δ=1 and $L=2\times 10^{4}$ for both panels. (**a**) Gradient *g* extracted from the data fitting S(L,l)∼gln[sin(πl/L)] of bipartite, steady-state EE vs. the subsystem size *l* at different loss rates for u=0.8. The vertical dashed line highlights the phase transition point at v=0.6. (**b**) The same gradient *g* as obtained in (**a**) vs. the real and imaginary parts of chemical potential μ=u−iv. The values of *g* at different (u,v) can be read out from the color bar.

**Figure 5 entropy-26-00272-f005:**
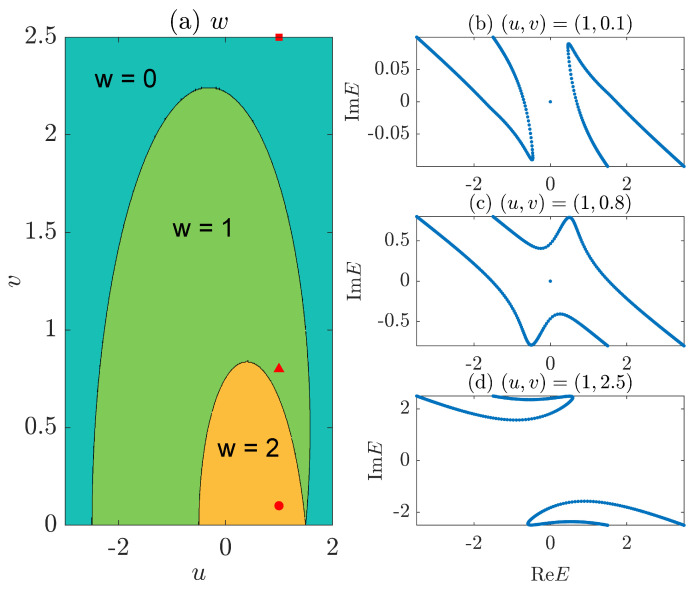
Topological phase diagram and typical spectra of the LKC with NNN hopping and pairing. (**a**) shows the winding number *w* vs. the real and imaginary parts of chemical potential *u* and *v*. The yellow, green and blue regions have w=2, 1 and 0, respectively. The red solid dot of (**a**) resides at (u,v)=(1,0.1), with the associated spectrum of ${\hat{H}}_{2}$ shown in (**b**). The red solid triangle of (**a**) is located at (u,v)=(1,1.1), and the associated spectrum of ${\hat{H}}_{2}$ is given in (**c**) on the complex energy plane. The red solid square of (**a**) lies at (u,v)=(1,2.5), and the associated spectrum of ${\hat{H}}_{2}$ is shown in (**d**). Other system parameters are $J_{1}=\Delta_{1}=1$ and $J_{2}=\Delta_{2}=1.5$ for all panels.

**Figure 6 entropy-26-00272-f006:**
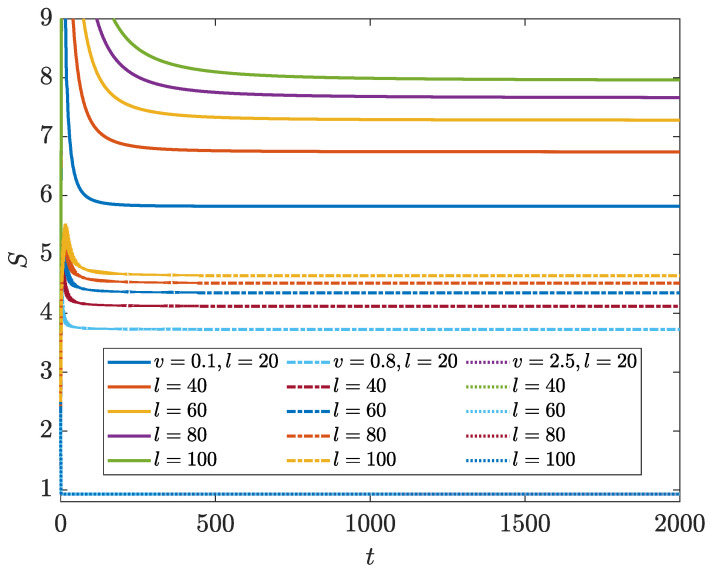
Bipartite EE vs. time *t* for the LKC with NNN hopping and pairing for the loss rate v=0.1 (solid lines), 0.8 (dash-dotted lines) and 2.5 (dotted lines). Other system parameters are $J_{1}=\Delta_{1}=1$, $J_{2}=\Delta_{2}=1.5$ and u=1. *l* denotes the subsystem size and the total lattice size is $L=2\times 10^{4}$.

**Figure 7 entropy-26-00272-f007:**
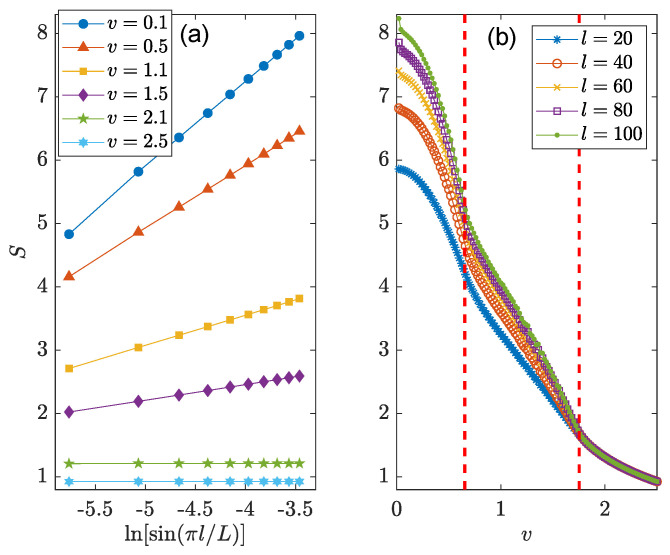
Bipartite EE of steady states vs. (**a**) the subsystem size *l*, and (**b**) the loss rate *v* for the LKC with NNN hopping and pairing. Other system parameters are $J_{1}=\Delta_{1}=1$, $J_{2}=\Delta_{2}=1.5$ and u=1 for both panels. The lattice size of the whole system is $L=2\times 10^{4}$. Vertical dashed lines in (**b**) denote two topological transition points of the system.

**Figure 8 entropy-26-00272-f008:**
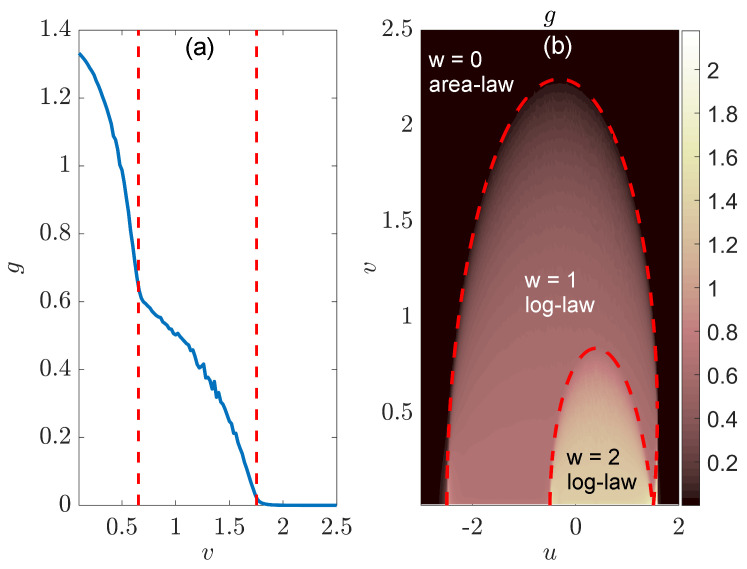
Entanglement phase transitions in the LKC with NNN hopping and pairing. System parameters are $J_{1}=\Delta_{1}=1$, $J_{2}=\Delta_{2}=1.5$ and $L=2\times 10^{4}$ for both panels. (**a**) Gradient *g* extracted from the data fitting S(L,l)∼gln[sin(πl/L)] of bipartite, steady-state EE vs. the subsystem size *l* at different loss rate *v* for u=1. Vertical dashed lines highlight topological transition points of the system. (**b**) The same gradient *g* as obtained in (**a**) vs. the real and imaginary parts of chemical potential μ=u−iv. The values of *g* at different (u,v) can be figured out from the color bar.

## Data Availability

Data are contained within the article.

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
