# Peer review of "Entanglement Phase Transitions in Non-Hermitian Kitaev Chains"

_entropy, 2024, doi:10.3390/e26030272_

Round 1

Reviewer 1 Report

Comments and Suggestions for Authors

The authors study the entanglement phase transition in non-Hermitian systems by the one-dimensional Kitaev chain.
Varying the value of the "loss rate", they found that the long-time behavior of the entanglement entropy of a subsystem characterizes the phase transition in the system, which shows that the dynamics of the entanglement entropy is a clue to identifying various types of topological phases.
The results of the manuscript are clearly presented and interesting, and I recommend the publication of the manuscript in Entropy.

Author Response

We thank our Reviewer 1 for the nice summary of our results and the recommendation for the publication of our manuscript in Entropy.

Reviewer 2 Report

Comments and Suggestions for Authors

The author studies entanglement phase transitions between phases of different topological winding number in non-hermitian Hamiltonians in symmetry class BDI in one spatial dimension (non-hermitian long-range Kitaev chains). 

Symmetry class BDI hosts Z-valued topological phases, where the topological quantum number, i.e., the winding number, describes the number of topologically protected Majorana edge states. Here, the conventional BDI Hamiltonian is modified by a complex chemical potential, making the Hamiltonian non-hermitian. Instead of the ground state, the author determines the stationary state of an evolution under such non-hermitian Hamiltonian. This stationary state is then explored in terms of its topological winding number and its entanglement entropy. Like the ground state of its hermitian counterpart, the stationary state of the non-Hermitian Hamiltonian can be obtained semi-analytically by utilising the Gaussian nature of free fermion states. Since no obstructions such as measurement-randomness or mixed states due to dissipation appear in this formulation, the analysis is straightforwardly executed. 

Overall the analysis of the non-hermitian Hamiltonian stationary states appears to be sound. It has not been performed for long-range models in class BDI and thus the work constitutes an original contribution to the field. I would be able to recommend publication. However, before that, two major concerns with the present work have to be resolved.

1.) A non-hermitian Hamiltonian formulation of dissipative or measurement dynamics has been considered recently quite often. In most cases the authors of such works consider the non-hermitian Hamiltonian as a simplification or a proxy for either dissipation, i.e., an open quantum system, or the evolution imprinted by measurement. The main reason is that the analysis of a static, non-Hermitian Hamiltonian is much more appealing than that of the full Lindblad master equation or the stochastic evolution due to measurement. However, in each such cases, the corresponding simplification that leads to the non-hermitian Hamiltonian need to be discussed. This is not the case in the present work. 

Let me stress this clearly: (i) Entanglement transitions due to measurement appear in wave functions that result from a stochastic evolution. Non-Hermitian Hamiltonians only appear in the so-called zero click or post-selected evolution, which discards a large set of potential measurement outcomes. (ii) A dissipative evolution induced by an environment leads to a mixed state (since the system becomes entangled with the environment) and thus follows a Lindblad master equation, which in addition is trace preserving. The non-Hermitian evolution is not norm preserving and thus can only be an approximation. The author needs to clarify which approximation underlies the use of a non-Hermitian Hamiltonian.

2.) The references provided by the author are incomplete. For entanglement transitions, many references to Clifford or Haar random circuits are provided but at the same time, many references to entanglement transitions in free fermion models are neglected. The same is true for references with respect to entanglement transitions in the same symmetry class as considered in this work. The author needs to improve on that.

Author Response

We thank our Reviewer 2 for the nice summary of our results and the identification that our work "constitutes an original contribution to the field". Our point-to-point responses to the questions and comments of our Reviewer 2 can be found in the uploaded PDF file.

Round 2

Reviewer 2 Report

Comments and Suggestions for Authors

The manuscript is improved. I would ask the author to also add the work 

SciPost Phys. 14, 031 (2023)

to the list of references. I recommend publication. Reasoning is given in the previous report.

Author Response

We have added the paper SciPost Phys. 14, 031 (2023) to our reference list (now the ref [59]). We thank our Reviewer 2 very much for the recommendation of our manuscript for publication.